# Physical Activity of Adolescents with and without Disabilities from a Complete Enumeration Study (*n* = 128,803): School Health Promotion Study 2017

**DOI:** 10.3390/ijerph16173156

**Published:** 2019-08-29

**Authors:** Kwok Ng, Päivi Sainio, Cindy Sit

**Affiliations:** 1School of Educational Sciences and Psychology, University of Eastern Finland, P.O. Box 111, FI-80101 Joensuu, Finland; 2Department of Physical Education and Sport Sciences, University of Limerick, V94 T9PX Limerick, Ireland; 3Department of Welfare, Finnish Institute for Health and Welfare, P.O. Box 30, FI-00271 Helsinki, Finland; 4Department of Sports Science and Physical Education, The Chinese University of Hong Kong, Hong Kong, China

**Keywords:** children, disability, teenager, Washington Group, functional difficulties, exercise

## Abstract

Evidence suggests that adolescent males take part in more moderate-to-vigorous physical activity (MVPA) than females, and that adolescents with disabilities participate in even less. Public health data are typically based on the international physical activity (PA) recommendations of at least 60 minutes of MVPA daily. However, it appears that data are lost because a person who reports MVPA 0–6 days a week is grouped together and is considered as ‘inactive’. Therefore, the purposes of this study were to report differences among adolescents with and without disabilities who were ‘active’ and ‘inactive’ and to explore differences by sex. A complete enumeration study (2017 School Health Promotion Survey; *n* = 128,803) of Finnish adolescents aged between 14–19 years old was conducted. The single item self-report MVPA was used with items from the Washington Group on Disability Statistics. Data were grouped into physiological and cognitive disabilities and were split into active and inactive adolescents based on the PA recommendations; subsequently, binary logistic regression analyses were performed. Data from the inactive participants were analyzed with multivariate analysis of covariance and effect sizes were reported. Approximately 10% of males and 17% of females reported disabilities. There were fewer adolescents with disabilities who took part in daily PA (OR = 0.90, CI = 0.85–0.94), especially among those with cognitive disabilities (OR = 0.86, CI = 0.82–0.91). There were more active male than female adolescents (OR = 1.48, CI = 1.43–1.52). Of the inactive adolescents, females reported similar MVPA to males, with and without disabilities after controlling for age, school type, and family financial situation. Inactive adolescents with walking difficulties reported the least amount of MVPA (males; mean = 2.24, CI = 2.03–2.44, females; mean = 2.18, CI = 1.99–2.37). The difference in means with adolescents without disabilities according to Cohen’s d effect size was medium for males (0.56) and females (0.58). The effect sizes from all other groups of disabilities were small. The difference in PA between males and females has diminished among the inactive groups, yet there is still a need to improve the gap between males and females, especially for those who meet the PA recommendations. More strategies are needed to improve MVPA among adolescents with disabilities, especially those with cognitive disabilities.

## 1. Introduction

Physical activity (PA) is a priority area in adolescent health promotion. Benefits in participation of PA can create improvements in the physical [1], mental [2], and social [3] health domains. In addition, regular PA habits during adolescence track into adulthood [4]. As such, there is worldwide interest to monitor PA levels during school-age years through national report cards [5]. The report card includes an indicator for governments to provide PA opportunities for all children and youth. This single indicator alone is considered, under a systems approach to PA promotion, to cover half of the factors linked with being physically active [6]. It is, therefore, not surprising that health promotion programs and legislation are considered to be the best methods for cost-effectiveness and return on investment in combatting physical inactivity [7].

Monitoring of key determinants can be used to gain awareness of the changes over time in public health and health promotion activities. At the time of compiling the recent global matrix of report cards from 49 countries [5], 47 of these countries have ratified the United Nations Convention on the Rights of Persons with Disabilities (CRPD). Countries with ratification of the CRPD indicated the implementation of the rights of persons with disabilities. According to the CRPD, Article 31—Statistics and data collection, “States parties undertake to collect appropriate information, including statistical and research data” and that the data “shall be disaggregated, as appropriate… to identify and address the barriers faced by persons with disabilities in exercising their rights” [8]. Despite the convention coming into force in 2008, few researchers conducted national statistics on PA behaviour, even though taking part in recreation, leisure and sport is considered a fundamental right under Article 30 of the CRPD [8]. In 2016, there was a call for future research into children and adolescent PA research to include disability data [9], yet in the subsequent Global Matrix, disability and chronic medical conditions appeared in only two countries [10,11]. 

It has been well argued that people with disabilities have the greatest need to be physically active, avoiding further complications to their existing disabling conditions [12]. As such, there is a large demand to capture information about PA levels among this population group. Such information can be used to inform policy makers, practitioners, researchers and individuals with disabilities. 

From a pragmatic perspective, “physically active” individuals are considered the people who meet the PA recommendations [13]. Previous research has demonstrated the need for children and adolescents aged between 5–17 years to participate in daily moderate-to-vigorous physical activity (MVPA) for at least 60 minutes to have a healthy life [13]. The same recommendations apply to children and adolescents with disabilities; however, “they should work with their health care provider to understand the types and amounts of PA appropriate for them considering their disability” [14]. According to the terminology used in these PA recommendations, active adolescents participate in daily MVPA, whereas all other adolescents are labelled as inactive. This may cause some confusion, because an adolescent who is active for six days a week and takes a day off for rest does not meet the PA recommendations and is therefore classified as inactive. This recommendation has been criticized by health promotion experts because increasing PA levels from zero days to daily might be harmful, with an associated greater risk of injuries [15]. Research evidence suggests that even some levels of PA are better than none [1]. Despite the message that something is better than nothing, few studies actually report the levels of PA among the inactive group of young adolescents. 

### 1.1. Interventions Targeting Female PA

Based on previous studies on child and adolescent PA levels, males are typically more physically active than females [16]. In Finland, the levels of PA fall more sharply between the ages of 11 years and 16 years among females compared with males [17]. There was a seven percent difference between males and females among active adolescents in Finland, despite being classified as a country with low inequality between males and females [18]. Many aspects in the lives of female adolescents differ from males [19]. For example, during this age, females tend to report more somatic symptoms of pain [20] and have lower perceptions of health [21], although they seem to outperform males in academic performance [22]. In addition, females with disabilities may participate in activities other than PA for their hobbies and are therefore less likely to adhere to PA recommendations [23]. These findings have led to several interventions with specific targets to improve female adolescents’ PA levels [24,25,26,27]. Although these studies have not included females with disabilities, it is only a matter of time before interventions target female adolescents with disabilities [28,29]. Most of these intervention studies are powered based on an increase in the intervention group of what is equivalent to 10% absolute difference in MVPA per day [30]. The studies in a review by Lonsdale et al. [30] do not advise for an immediate shift of adolescents who go from a few days of 60 minutes of MVPA to daily PA. As such, it is important to report national data on PA levels of the inactive group. 

### 1.2. Purpose of the Study

Given the gaps in the scientific literature in reporting PA levels of the inactive groups of adolescents and low reporting of national data of PA levels among adolescents with disabilities, the purposes of this study were (1) to report the PA levels of Finnish adolescents after disaggregation by disability, (2) to investigate the prevalence of physically active males and females after disaggregation by disability, and (3) to examine the differences in PA between the inactive males and females after disaggregation by disability through the Washington Group on Disability statistics [31]. The questionnaire set was used to ensure consistent reporting of disabilities for international studies based on the CRPD [32].

## 2. Materials and Methods 

### 2.1. Data

The 2017 School Health Promotion Study was a self-report nationwide survey of children aged between 10–12 years old and adolescents aged between 14–20 years old in Finland, with a total reach of 240,320 children and adolescents across the country. For this study, only data from adolescents were used (*n* = 142,350). The study was based on data collected on the total population (complete enumeration), and each municipality decided whether schools would participate in the survey. Four out of 320 municipalities decided no schools would participate and 21 municipalities either did not respond or did not have a general upper secondary school [33]. The response rate at the respondent level ranged from 64% for adolescents in general upper secondary school (typical age 14–16 years) and estimates of 50% for adolescents in high school (typical age 16–19 years) and 40% for adolescents in vocational school (typical age 16–19 years) [34]. Data were cleaned so that it was possible to carry out the analyses where the final sample size was *n* = 128,803, reduced through a stepwise reduction based on missing age (*n* = 1175), those outside of the targeted age range between 14–19 years (*n* = 3280), missing sex information (*n* = 661) and lack of disability data (*n* = 3685). Where respondents reported difficulties in all domains of functional difficulties, the data were treated as spoilt and removed (*n* = 612). Data were collected from 1st March until 31st May 2017. Ethical approval for carrying out the data collection was through the Ethics committee of the National Institute for Health and Welfare, Finland. The study abided by the international (Helsinki declaration) and national requirements for consent, assent and voluntary and anonymous participation. It was confidential and was administered in the classroom. For more information about the study and access to the datasets, please visit https://thl.fi/en/web/thlfi-en/research-and-expertwork/population-studies/school-health-promotion-study.

### 2.2. Measures

Demographic variables included sex, age from a calculation based on when the survey was completed and the given month and year of birth from the respondents. Data were stratified by school type—general upper secondary schools, high schools or vocational schools. There was an item used as a proxy for parental social economic status (SES) through the question, “how would you rate your family’s financial situation?” with optional responses of “very good”, “fairly good”, “moderate”, “fairly poor”, and “very poor”. 

#### 2.2.1. Physical Activity

A self-reported item from the PACE+ PA questionnaire was used [35]. Respondents had to recall over the past seven days their frequency, intensity and duration of MVPA. In the survey, there was a scripted description of MVPA with some examples of where MVPA takes place and what the MVPA was. The back translation from Finnish included the question: “Think about all the moving around you have done over the past 7 days. On how many days have you been on the move for at least one hour per day?” The response category range was between 0 to 7 days. This item has been reported to have acceptable validity [36] and reliability [37], hence its suitability for surveillance data [38]. 

Data were recoded in two ways. The first was to account for active individuals (reported 7 days) or inactive (6 or fewer days) as suggested by the international PA recommendation [14]. The inactive group was then analyzed as a separate group with a range from 0 to 6 days of MVPA for the multiple linear regression analyses.

#### 2.2.2. Disabilities

Items related to functional difficulties were used to generate a marker for disabilities based on the questions of the Washington Group on Disability Statistics (WG) [32]. For children and adolescents, WG together with UNICEF created a child functioning module (CFM) consisting of 24 questions intended for parents or primary caregivers to respond to [39]. The School Health Promotion Survey is solely self-report by adolescents, hence the items were slightly modified to reflect this. The terminology in original questions were subsequently modified from proxy items (“does [child’s name] have difficulties in…”) to self-report, whereby the respondent answered questions beginning with “do you have difficulties in…” The functions from the CFM were used, but due to the space limits in our questionnaire, six items were chosen covering the functions of seeing, hearing, walking, remembering, learning and concentrating. The questions had a four-point response scale to describe the intensity of difficulties; “Not difficult at all”, “A little difficult”, “Very difficult”, and “I cannot do it at all”. 

Disability was determined as having “very difficult” or “cannot do” in any of the six activities and were dichotomized into disability versus no disabilities when the reporting was “not difficult at all” or “a little difficult”. Then, two macro functioning groups were formed by grouping the physiological functions (seeing, hearing, walking), and the cognitive functions (remembering, learning, concentrating) together. 

### 2.3. Analyses

Binominal logistic regressions were used to determine odds ratios, with 95% confidence intervals, of meeting the PA recommendation with disabilities being the independent variables after adjusting for sex, age, family financial situation and school type. Subsequent analyses were carried out on the associations between disabilities and PA of the individuals who were grouped as inactive. Tests of normality were carried out on the distribution of PA levels after stratifying by sex. After meeting the assumptions of normality by sample skewness and kurtosis, ANCOVAs were repeated with disability groups and school types and with age and family financial situation as covariates. The use of ANCOVA was used because the covariates of age and financial situation are known to be associated with PA; however, for this study these covariate effects were not explored further. Differences in the adjusted days of MVPA between disability groups and with no disabilities were tested. A conversion of standard deviation of the adjusted mean number of days was used by the 95% confidence intervals to then be used to measure effect sizes from Cohen’s d. Measurement of effect sizes through differences of means were used to determine the practical significance in addition to the *p*-values given from the statistical significance [40]. Interpretations of these effect sizes were based on Cohen’s d interpretation that values between 0.20–0.49 were small, 0.50–0.79 were medium and over 0.80 were large [41].

## 3. Results

### 3.1. Descriptive Results

Approximately 10% of males (*n* = 6385) and 17% of females (*n* = 11,107) reported difficulties in at least one functional difficulty, and the estimated prevalence of disability among adolescents aged between 14–19 years old in Finland was 13.5% (Table 1). Over half the adolescents attended general upper secondary school (54%). There were more females in high school (30%) compared with vocational schools (16%), whereas among males the figures were more even (22% vs. 24%). Fewer females (61%) reported their family’s financial situation was either very or fairly good than males (70%). The distribution of PA by days can be found in Appendix A. 

### 3.2. Active Populations

Almost one in five males (19.3%) reported participation in daily MVPA, whereas significantly fewer females (13.3%) reported the same frequency of behaviour (Table 1). This was also true (OR = 1.48, CI = 1.43–1.52) after adjusting for age, school type, and family financial situation (Table 2). Moreover, as age increased from 14 years old to 19 years old, there were fewer adolescents who reported daily MVPA. In addition, when compared to attendance in general upper secondary school, fewer adolescents in vocational school and high school reported daily MVPA. The association with a family’s financial situation was not linear: When compared to adolescents who considered their families very poor, participation in daily MVPA was more uncommon among those with fairly poor or moderate financial situations. However, daily MVPA was more common among those with a very good financial situation (Table 2).

After adjusting for sex, age, school type and SES, there were fewer active adolescents with disabilities than adolescents without disabilities (Table 2). More specifically, there were fewer active adolescents with walking difficulties, remembering difficulties, learning difficulties, concentrating difficulties or general cognitive difficulties than their peers without disabilities. The associations between daily MVPA and seeing and hearing were not statistically significant.

### 3.3. Inactive Populations

After excluding the study participants who reported daily MVPA, inactive (six or fewer days of MVPA) females reported more days of MVPA than inactive males (*p* < 0.001) (Table 3). This pattern was confirmed for those aged 15 years (*p* = 0.001), 16 years (*p* < 0.001), and 17 years (*p* = 0.038) old, for individuals with very poor *(p* = 0.004), moderate (*p* < 0.001), fairly good (*p* < 0.001) and very good (*p* < 0.001) family financial situations. In addition, there were sex differences in the days of MVPA among adolescents without disabilities (*p* < 0.001), with disabilities (*p* < 0.001), with specifically hearing (*p* < 0.001), remembering (*p* < 0.001), learning (*p* < 0.001) and concentrating difficulties (*p* = 0.003). The effect sizes for all the comparisons were smaller than 0.2 and were interpreted to be less than small [41]. 

For all disability groups, the adjusted levels of PA, after controlling for age, SES and school type among inactive adolescents, both males and females, were lower than adolescents without disabilities (Table 4). The largest differences were between adolescents with walking difficulties and without disabilities and the effect size was medium for both males (d = 0.56) and females (d = 0.58). The effect size for inactive adolescent females with any disabilities was below the threshold for small (d = 0.17) when compared to inactive females without disabilities, whereas for males the effect size was small (d = 0.22). The difference in PA among males with seeing difficulties was also below the threshold for small (d = 0.13), as was the effect sizes for females with seeing (d = 0.12), hearing (d = 10), physiological disabilities combined (d = 0.15) and for cognitive disabilities combined (d = 0.19). 

## 4. Discussion

### 4.1. Main Findings from the Study

There were two main findings in this study. First, fewer adolescents with disabilities took part in daily MVPA after controlling for sex, age, school type and SES, and overall the levels of PA were lower among the inactive group. Previous studies often report that people with disabilities are less physically active than their peers without disabilities [42], although we found the degree of lowered PA varies depending on functional disabilities, with adolescents with walking difficulties the least active. The second main finding was that although there were more males who reported to meet the PA recommendations, among adolescents who did not meet the PA recommendations (inactive group), females were as active as inactive males. The results of this study were similar to previous national [23,43] and international studies [5], where more males meet the PA recommendations than females, although in this paper we provide greater insight through reporting levels of PA among inactive adolescents. These findings are discussed further.

This study examined the associations of PA with each disability group. Of the physiological difficulties group, only walking difficulties were negatively associated with taking part in daily MVPA. Similarly, the effect size of the inactive walking difficulties group was medium when compared to adolescents without disabilities, but was small for males with hearing difficulties and less than small for males and females with seeing difficulties. In other words, the extent of the differences in daily MVPA between adolescents with seeing or hearing difficulties and the general population were negligible. Another consideration for interpretation is that the measure used from the WG for walking difficulties is a crude indicator for people with physical impairment or disabilities [32]. Researchers have repeatedly found that adolescents with physical impairment or disabilities have difficulty in (1) accessing facilities to be physically active, (2) finding suitable activities to do, (3) having friends to do the activities with, and (4) having the self-confidence to do such activities [44,45]. With many barriers, both personal and environmental, it is not surprising that one and a half times as many adolescents with walking difficulties do not meet the daily PA recommendations when compared with their peers without disabilities. 

There were fewer active adolescents with cognitive disabilities than peers without disabilities and this is commonly cited in the literature [46]. For example, overall PA time depends on the severity of the cognitive impairment among males [47]. In addition, adolescents with cognitive difficulties encounter other environmental barriers, such as being able to communicate with people when taking part in physical activities [48]. Programs like the Special Olympics Unified Sports Programme have been effective in making the organized sport environment a safe one [49]. The Unified Sports program is based on the inclusion of athletes with and without intellectual disabilities playing with and against each other. In 2015, there were 1860 registered athletes and unified partners in the Finnish Special Olympics database, with a target to increase year on year [50]. Such programmes can be an important link for health promotion activities, which would increase the levels of PA with the aim to get more adolescents in the ‘active’ group. 

Based on previous studies, researchers suggest the need for PA promotion strategies targeting females because they are considered an inactive population [16,51]. Yet, researchers also highlighted reaching daily MVPA is too ambitious for inactive people, hence health promotion activities require individuals to set realistic targets that are more suitable and sustainable, while still providing health benefits [15]. Through the examination of the inactive group, females reported slightly more days of MVPA than inactive males, but the effect sizes were trivial. Nonetheless, it is rare to see levels of PA that are similar between males and females [16,51]. This novel finding may be interpreted as a success of the health promotion activities that were successful among female adolescents. Across the population. Since 2010, the Finnish Schools on the Move study has been driving whole of school PA promotion, whereby intervention effects were stronger for females than males [52]. There have also been recent initiatives from Finnish sports council to increase the levels of PA among females [53]. In addition, between 1985 and 2014 there was a 10% increase among 15 years old females who took part in organized sport clubs [54]. Furthermore, the rise in males who take part in computer gaming has decreased the low levels of PA among the inactive population further [55], and may have made the gap between those that are active and inactive wider. In addition, behavioural compensation such as engagement with high levels of risk behaviours among adolescents whereby physical fighting are more common among males than females [18] need to be taken into account for explaining how the averages of MVPA were not different between males and females. Other male targeting behavior interventions may also need to consider increasing levels of PA as the gap between ‘active’ and ‘inactive’ widens in this population.

The participants in this study were from 14 to 19 years old and were across three different school environments. Response rates were highest among the secondary schools and lowest among the vocational schools. There were some difficulties to collect data among adolescents aged between 16–19 years old, particularly those attending vocational schools. Vocational school curriculum can include spending just a few hours a day on site. As part of a complete enumeration study, survey completion was limited to a specific time frame. Therefore, when students were out on field work placement or absent in other ways, they could not complete the survey. Despite this limitation, there were over 25,000 respondents in the vocational school setting from this study and data were collected at random [34]. The low levels of PA have been recognized by other researchers and there is a great need to carry out PA promotion activities among students in vocational schools [56]. In this study, fewest students in vocational schools reported daily MVPA and the inactive students in vocational schools reported, on average, half a day less of MVPA than students in secondary schools. The results corroborate with previous findings, whereby adolescents with disabilities in general upper secondary schools with low levels of PA were associated with aspirations to go to vocational schools rather than high schools [45]. As such, programs like those from Hankonen et al. [56] attempted to increase the levels of PA among adolescents in vocational schools and they require urgent examination. Further consideration of frameworks like the International Classification of Functioning, Disability and Health [57], and applying adapted physical activity practices is necessary because there is a increase likelihood that within vocational schools there would be students with disabilities. 

### 4.2. Study Limitations

The interpretation of the results of this study is specific to Finnish adolescents and requires the consideration of some study limitations. The measures used in this study were self-reported. We excluded reports from the easy-to-read questionnaire because the answering categories of functional difficulties were dichotomized and therefore not comparable with the 4-point response scale. There was a lack of testing of the PACE+ item among children with moderate to severe disabilities. Therefore, there may be some reporting bias based on the interpretation of the questions asked. Yet, the survey was administered in the same way to all and hence, we feel that with the sample size, despite not being weighted, such anomalies may be smoothed out by the overall number of respondents. Moreover, there were no variables available to use for class- or school-based interaction effects and this may have influenced the results, but to a lesser extent as a complete enumeration study than other national representative studies using proportion-to-size ratio sampling. Lastly, there are multiple determinants of PA participation, and not all confounding variables were included in the analyses because of the availability of variables and the nature of secondary analyses from the dataset. We chose those variables that have been highlighted strongly in the literature as confounding, namely, age, SES and school type, to adjust the findings. More studies are needed to examine other effects such as participation in organized and unorganized activities. 

## 5. Conclusions

The majority of males with disabilities reported lower levels of PA than their same sex peers without disabilities, while for females the difference was not as concrete. In addition, fewer adolescents with cognitive disabilities or walking difficulties met the PA recommendations than adolescents without disabilities. The differences in PA levels between males and females remained apparent, with 40% higher chance for males than females in taking part in the recommended amount of MVPA in a week. Of the adolescents who did not meet the PA recommendations for health (inactive group), the females reported as much PA as males. More interventions are required to reduce the gap in females to meet the active criteria, and the gap in PA levels among inactive adolescents could be emphasized on population groups of individuals with cognitive disabilities, walking difficulties, or those attending vocation schools. 

## Figures and Tables

**Table 1 ijerph-16-03156-t001:** Population descriptive statistics by sex.

Variables	*N*	Male % 62,746	Female % 66,057	Chi-Square *p*
Total	128,803	48.7	51.3	
Type of School				<0.001
General Upper Secondary	69,342	53.6	54.0	
High School	33,904	22.3	30.1	
Vocational	25,557	24.0	15.9	
Age				<0.001
14	23,016	17.5	18.3	
15	33,729	26.0	26.4	
16	31,852	25.1	24.4	
17	27,641	22.0	21.0	
18	9916	7.8	7.6	
19	2649	1.6	2.4	
Family financial situation				<0.001
Very poor	1480	1.1	1.3	
Fairly poor	7840	4.9	7.5	
Moderate	33,724	23.9	29.9	
Fairly good	55,675	44.9	44.5	
Very good	25,930	25.2	16.8	
Disabilities				
*No disabilities*	111,311	89.9	83.2	REFERENCE
*Disabilities*	17,492	10.2	16.8	<0.001
Seeing difficulties	4107	2.8	4.3	<0.001
Hearing difficulties	1615	1.4	1.5	0.171
Walking difficulties	1042	1.0	0.9	0.009
*Physiological disabilities*	5800	4.0	5.9	<0.001
Remembering difficulties	9308	5.5	9.8	<0.001
Learning difficulties	6386	3.6	7.2	<0.001
Concentrating difficulties	2985	1.7	3.5	<0.001
*Cognitive disabilities*	13,669	7.8	13.9	<0.001
Moderate to Vigorous Physical Activity				<0.001
Not daily	107,945	80.7	86.7	
Daily	20,858	19.3	13.3	

**Table 2 ijerph-16-03156-t002:** Adjusted binary logistic regressions for meeting physical activity recommendations.

Variables	OR	LCI	UCI
Sex			
Male	**1.48**	**1.43**	**1.52**
Female	1.00 (Ref)		
Age			
14	1.00 (Ref)		
15	**0.78**	**0.75**	**0.81**
16	**0.75**	**0.71**	**0.79**
17	**0.70**	**0.65**	**0.75**
18	**0.63**	**0.57**	**0.69**
19	**0.75**	**0.66**	**0.87**
School Type			
Secondary	1.00 (Ref)		
High	**0.80**	**0.75**	**0.85**
Vocational	**0.72**	**0.67**	**0.77**
Family Financial Situation			
Very poor	1.00 (Ref)		
Fairly poor	**0.71**	**0.61**	**0.83**
Moderate	**0.77**	**0.67**	**0.89**
Fairly good	0.89	0.77	1.03
Very good	**1.42**	**1.23**	**1.64**
Disabilities			
No disabilities	1.00 (REF)		
*Disabilities*	**0.90**	**0.85**	**0.94**
Seeing difficulties	1.04	0.95	1.14
Hearing difficulties	1.09	0.95	1.25
Walking difficulties	**0.67**	**0.55**	**0.82**
*Physiological disabilities*	1.00	0.92	1.07
Remembering difficulties	**0.88**	**0.83**	**0.94**
Learning difficulties	**0.90**	**0.84**	**0.98**
Concentrating difficulties	**0.77**	**0.69**	**0.87**
*Cognitive disabilities*	**0.86**	**0.82**	**0.91**

**Table 3 ijerph-16-03156-t003:** Association of covariates and unadjusted days of physical activity of inactive group.

Variables	Males		Females		Total		*t*-Test *	Effect Size *
	Inactive PA		Inactive PA		Inactive PA			
	mean	SD	mean	SD	mean	SD	*p*	d
Total	3.33	1.81	3.39	1.74	3.36	1.77	<0.001	−0.03
Type of School								
General Upper Sec	3.51	1.77	3.56	1.68	3.54	1.72	<0.001	−0.03
High School	3.40	1.80	3.37	1.76	3.38	1.77	0.135	0.02
Vocational	2.91	1.82	2.90	1.77	2.90	1.80	0.711	0.01
Age								
14	3.72	1.72	3.72	1.64	3.72	1.68	0.858	0.00
15	3.45	1.77	3.52	1.69	3.48	1.73	0.001	−0.04
16	3.27	1.81	3.36	1.74	3.32	1.78	<0.001	−0.05
17	3.17	1.81	3.22	1.77	3.19	1.79	0.038	−0.03
18	3.00	1.85	3.07	1.78	3.03	1.81	0.062	−0.04
19	2.80	1.89	2.77	1.81	2.78	1.84	0.741	0.02
Family Financial Situation								
Very poor	2.53	2.07	2.85	1.83	2.71	1.95	0.004	−0.17
Fairly poor	2.94	1.86	2.96	1.81	2.95	1.83	0.791	−0.01
Moderate	3.15	1.82	3.23	1.75	3.19	1.78	<0.001	−0.04
Fairly good	3.40	1.77	3.50	1.69	3.45	1.73	<0.001	−0.06
Very good	3.55	1.79	3.69	1.70	3.61	1.75	<0.001	−0.08

* Sex difference with males as reference.

**Table 4 ijerph-16-03156-t004:** Differences in adjusted means, 95% confidence intervals of moderate-to-vigorous physical activity (MVPA) days by sex with effect sizes with no disabilities as comparison group.

Variables	Male Inactive PA Days				Female Inactive PA Days			
	Mean	LCI	UCI	d	Mean	LCI	UCI	d
No Disabilities	3.37	3.35	3.39	REF	3.40	3.38	3.42	REF
Disabilities	2.93	2.87	2.98	0.22	3.04	3.00	3.07	0.17
Seeing	3.11	3.00	3.22	0.13	3.15	3.07	3.23	0.12
Hearing	2.90	2.73	3.07	0.23	3.21	3.07	3.34	0.10
Walking	2.24	2.03	2.44	0.56	2.18	1.99	2.37	0.58
Physiological	2.93	2.84	3.02	0.22	3.09	3.02	3.15	0.15
Remembering	2.90	2.82	2.97	0.33	2.97	2.92	3.02	0.29
Learning	2.75	2.65	2.85	0.31	2.92	2.86	2.99	0.23
Concentrating	2.72	2.59	2.85	0.33	2.86	2.78	2.95	0.26
Cognitive	2.88	2.81	2.94	0.24	3.00	2.96	3.04	0.19

Means adjusted after controlling for age, SES and school type. LCI–Lower 95% confidence interval, UCI–Upper 95% confidence interval, d–Cohen’s d.

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
