# Peer review of "Physical Activity of Adolescents with and without Disabilities from a Complete Enumeration Study (n = 128,803): School Health Promotion Study 2017"

_ijerph, 2019, doi:10.3390/ijerph16173156_

Round 1

Reviewer 1 Report

Thank you for inviting me to review this manuscript that examined differences in MVPA between adolescents with and without disabilities and the modifying effects of sex. This manuscript makes use of a large sample size and is generally well written. There are some major concerns regarding the data analytic plan and the reporting of the results that need to be addressed. My specific comments are attached. 

Abstract:

-Please rephrase the purpose statement in the abstract to improve clarity. 

-Pertinent mean differences, Inferential statistics, and effect sizes should be reported within the abstract.

-Why was cognitive disabilities mentioned in the concluding sentence when it was not mentioned in the methodology presented in the abstract?

Introduction:

-Line 62: Reference is needed here. 

-Lines 83-84: This statement is not backed by empirical evidence. Please consider revising this statement. 

-Line 95: Please explicitly state the gaps in the current literature that this study aims to fill. 

-Lines 98-101 would be more appropriate in the Methods section, not at the end of the Introduction section. 

Methods:

-Were data missing at random?

-Were there attempts to reduce the possibility of social desirability bias?

-Were the data weighted? If yes, was this incorporated in the analysis?

-There is clustering within the data structure, how was this accounted for? There is no mention of the use of random effects. 

-For the ANCOVA, was the assumption of homogeneity of regression slopes met? If not, a general linear mixed effects model controlling for age is recommended. 

Results:

-Table 3: Use of 3 decimal places is excessive, 2 is recommended.

-Table 3: Please bold or flag significant parameter estimates.

-Please reported significant mean differences in text for the reader. 

-Where are the effect sizes? These need to be reported in text or in the tables. 

Discussion:

-Please comment on the magnitude (effect size) of these mean differences. 

-What other effect modifiers could have influenced the results?

-Please comment on the internal validity of these findings and the extent to which these results can be generalized. 

-A much better concluding sentence is needed to efficiently and effectively summarize the observed results. 

Author Response

Thank you for inviting me to review this manuscript that examined differences in MVPA between adolescents with and without disabilities and the modifying effects of sex. This manuscript makes use of a large sample size and is generally well written. There are some major concerns regarding the data analytic plan and the reporting of the results that need to be addressed. My specific comments are attached. 

We would like to thank the reviewers for the time to perform the review and examining the manuscript. The comments have helped the authors reconsider how to present the results, discussions of the paper after a reanalyses based on the reviewers comments.

Abstract:

-Please rephrase the purpose statement in the abstract to improve clarity. 

Thank you for the suggestion. We have stated the abstract purpose statement with more precision.

-Pertinent mean differences, Inferential statistics, and effect sizes should be reported within the abstract.

The OR, Mean differences and ES in the abstract.

-Why was cognitive disabilities mentioned in the concluding sentence when it was not mentioned in the methodology presented in the abstract?

We have included the classification groups from the Washington group to include cognitive disabilities.

Introduction:

-Line 62: Reference is needed here. 

Reference has been included.

-Lines 83-84: This statement is not backed by empirical evidence. Please consider revising this statement. 

We have removed this sentence as we realise it is quite irrelevant to the paragraph.

-Line 95: Please explicitly state the gaps in the current literature that this study aims to fill. 

We have included a statement in this line to clarify the gaps.

-Lines 98-101 would be more appropriate in the Methods section, not at the end of the Introduction section. 

WE have modified this sentence so that it still fits in the introduction as we would like the reader gain some background information as to our concept of disability. We do describe it further in the methods section.

Methods:

-Were data missing at random?

We can confirm that missing data from age, outside age range, missing sex information, and lack of disability data did not significantly differ in physical activity levels, and can therefore be considered to be random.

Chi-square test of independence of daily PA

Missing age p=0.750

Missing sex p=0.282

-Were there attempts to reduce the possibility of social desirability bias?

This was reported in the study limitations on line 277-8.

-Were the data weighted? If yes, was this incorporated in the analysis?

This was a complete enumeration study that did not use weights and is stated as a study limitation.

-There is clustering within the data structure, how was this accounted for? There is no mention of the use of random effects. 

The reviewer is correct in that there are potential clustering effects, however because this is secondary analyses, data at the class level were not available to perform multi-level analyses. This is a limitation that we stated on lines 280-1.

-For the ANCOVA, was the assumption of homogeneity of regression slopes met? If not, a general linear mixed effects model controlling for age is recommended. 

The reviewer is correct in the pure statistical way of using ANCOVA and the assumption of homogeneity of regression slopes. However, the ANCOVA was used instead of a generalised linear model because we are not looking to examine the impact of the covariates on the outcome variable and both tests, although calculated differently, provide the same results. The key difference, according to Grace-Martin, a statistician tutor, is the interest on the interaction between the variables. For us, we do not want to focus on the covariates in the linear model. https://www.theanalysisfactor.com/general-linear-model-anova-regression-same-model/

Results:

-Table 3: Use of 3 decimal places is excessive, 2 is recommended.

We have modified to 2dp as suggested by the reviewer.

-Table 3: Please bold or flag significant parameter estimates.

We have modified to put in bold

-Please reported significant mean differences in text for the reader. 

We have included more explicit reporting of the results for Table 3 and 4.

-Where are the effect sizes? These need to be reported in text or in the tables. 

The reviewer has made a really useful suggestion and we have reported the effect sizes.

Discussion:

-Please comment on the magnitude (effect size) of these mean differences. 

We have included some sentences to comment on the effect size in the discussion section.

-What other effect modifiers could have influenced the results?

We agree that our study is limited by the few effect modifiers in our study this is reported in limitations section.

-Please comment on the internal validity of these findings and the extent to which these results can be generalized. 

We have described the internal validity of these findings through the limitations section of ‘smoothing’. We would rather not describe the generalisability of the results as there are descriptive reports based on national data. Each country is different and we state that in the study limitations.

-A much better concluding sentence is needed to efficiently and effectively summarize the observed results. 

We have now re-written the conclusion section altogether.

Reviewer 2 Report

Lack of detailed information about the survey questionnaire used for the research (eg. subject of questions, reliability of the Cronbach test)

No information about the number of parents being examined, which the authors write about in chapter 2.2.2.

No verification of the assumed hypothesis.

No page number for citations (lines 52 and 67).

Under the tables there is no source quoted, e.g. your own research, your own elaboration.

Conclusions are clearer if they are punctuated.

The last sentence in chapter 5 is not a conclusion.

Pre-publication the text requires additions.

Author Response

Lack of detailed information about the survey questionnaire used for the research (eg. subject of questions, reliability of the Cronbach test)

We would like to politely refer the review to lines 139 for the PA question and 145-148 for the disability items. The PA question is a single item, and the functional difficulty items relate to different functions, that can be independent from each other. Therefore, we do not believe a Cronbach test is suitable for testing.

No information about the number of parents being examined, which the authors write about in chapter 2.2.2.

We did not ask parents. We have added a statement that this was solely self-report.

No verification of the assumed hypothesis.

We do not state hypotheses in the purpose of the study.

No page number for citations (lines 52 and 67).

In this referencing system, there is no room to include page number for citations. Please also not, the reference belongs to the UN convention, and the following reference can be found on page S364 for reference 9.

Under the tables there is no source quoted, e.g. your own research, your own elaboration.

We believe in this journal that all tables were from our own data and annotations are only used for providing further explanation for abbreviations.

Conclusions are clearer if they are punctuated.

We are not sure if this is possible with the journal

The last sentence in chapter 5 is not a conclusion.

We agree with the review and the sentence has been moved.

Pre-publication the text requires additions.

We are not sure we understand this reviewer’s comment.

Reviewer 3 Report

Major comments

1.  Active is defined for 7 days in past 7 days, and inactive is defined for 6 or fewer days. The definition of these categories is not sufficiently valid. In particular, this cut-off could influence sex-difference analysis.

2. The sex-difference is not so large. The difference in OR almost only less than 0.1, which might be meaningless for practical and real world. I am interested in the distribution from 0 to 7 days. Also, the sex-difference was hardly discussed. Therefore, it is possible to delete sex-difference in the manuscript.

Minor comments

3. Line 164-173, P4 and Table 1: Since I am not familiar with statistics, I could not understand the statistics. Also, why the number of subjects were difference Table 1 and Table 2.

4. Table 3: The title should state for which OR were, and OR was adjusted.

5.Table 4: It should be stated that the unit of figures in the table is day. The age was not need decimal point.

6. Table 5: I am sorry that I could not understand, especially different from Table 4.

Author Response

Lack of detailed information about the survey questionnaire used for the research (eg. subject of questions, reliability of the Cronbach test)

We would like to politely refer the review to lines 139 for the PA question and 145-148 for the disability items. The PA question is a single item, and the functional difficulty items relate to different functions, that can be independent from each other. Therefore, we do not believe a Cronbach test is suitable for testing.

No information about the number of parents being examined, which the authors write about in chapter 2.2.2.

We did not ask parents. We have added a statement that this was solely self-report.

No verification of the assumed hypothesis.

We do not state hypotheses in the purpose of the study.

No page number for citations (lines 52 and 67).

In this referencing system, there is no room to include page number for citations. Please also not, the reference belongs to the UN convention, and the following reference can be found on page S364 for reference 9.

Under the tables there is no source quoted, e.g. your own research, your own elaboration.

We believe in this journal that all tables were from our own data and annotations are only used for providing further explanation for abbreviations.

Conclusions are clearer if they are punctuated.

We are not sure if this is possible with the journal

The last sentence in chapter 5 is not a conclusion.

We agree with the review and the sentence has been moved.

Pre-publication the text requires additions.

We are not sure we understand this reviewer’s comment.

Reviewer 3

Major comments

1.  Active is defined for 7 days in past 7 days, and inactive is defined for 6 or fewer days. The definition of these categories is not sufficiently valid. In particular, this cut-off could influence sex-difference analysis.

We would like to politely disagree with the reviewer in that the definition are crude and are used commonly in public health. We do understand that it is not makes very little sense to place a person who is active for 6 days to be in the same category of 3 days, let alone 0 days, but based on the prior evidence that was used to build up the physical activity for health recommendations, the cut off is currently 7 days. This is the premise for the definition for ‘active’ and ‘inactive’, and it is commonly used in the scientific literature.

2. The sex-difference is not so large. The difference in OR almost only less than 0.1, which might be meaningless for practical and real world. I am interested in the distribution from 0 to 7 days. Also, the sex-difference was hardly discussed. Therefore, it is possible to delete sex-difference in the manuscript.

We are not sure where the reviewer found the OR to be less than 0.1. In Table 3, we have stated the OR for 7 days for Males is 1.475, with CI between 1.43-1.52. We would like to disagree with the reviewers comments to say it is not meaningful for practical, real world and scientific purposes.

The reviewer also stated their own interest in the distribution from 0-7 days. We believe we have presented this in Table 3, but using the international recommendation cut off from 0-6 days (reference) and 7 days (line 140-1).

The reviewer made a good suggestion to remove the reference to sex differences as the effect sizes were too small to warrant reporting of differences. However, the fact that there were not differences is actually very important to our field in terms of promotion of physical activity among girls.

We have incorporate further analyses of differences between disability and non disabilities, and observing the lack of differences between sexes.  

Minor comments

3. Line 164-173, P4 and Table 1: Since I am not familiar with statistics, I could not understand the statistics. Also, why the number of subjects were difference Table 1 and Table 2.

Thank you for the reviewer to notice the differences between Table 1 and Table 2. We have corrected the numbers represented in Table 1 for the ‘inactive’ only group and labelled the table accordingly.  

4. Table 3: The title should state for which OR were, and OR was adjusted.

We would like to thank the reviewer for this comment, and we have modified the title for Table 3.

5.Table 4: It should be stated that the unit of figures in the table is day. The age was not need decimal point.

We have modified the title of the Table to include ‘day’ and removed the decimals from age.

6. Table 5: I am sorry that I could not understand, especially different from Table 4.

We have removed this table.

Round 2

Reviewer 1 Report

-In the abstract, please indicate the type of effect size used on line 33. If this is Cohen’s d , these are medium effect sizes not large effect sizes (as you report on line 226).

-Because of the research design, please replace the words “increased” and “decreased” with “higher” and “lower”, respectively.

Author Response

The authors would like to kindly thank the reviewer to carry out a 2nd review of the manuscript. 

We have reviewed the comments and respond to each one of them.

-In the abstract, please indicate the type of effect size used on line 33. If this is Cohen’s d , these are medium effect sizes not large effect sizes (as you report on line 226).

Thank you for noticing the differences between the abstract and the results. We have modified it accordingly and included the inclusion of 'Cohen's d' in the sentence. This is in red colour font.

-Because of the research design, please replace the words “increased” and “decreased” with “higher” and “lower”, respectively.

We have made changes to L331 so that higher is replacing increase. We could not find anywhere in the document where decreased/decreases/decrease appears in the document that could be modified to 'lower'. This is in red colour font. 

Reviewer 2 Report

I have no comments for the authors

Author Response

The authors would like to thank the reviewer for supporting the manuscript and not adding further comments. 

Reviewer 3 Report

Again, I recommend that that the authors should reconsider the analysis of inactive groups. Simply, the question is which is more active males or females? The proportion of PA recommendation was higher in males, but among inactive groups females showed higher PA days. I think that the distribution of PA day should be shown and PA days could be compared for not inactive groups but for all study subjects.

Concerning my former comment 2, small OR difference (less than 0.1) means the difference in OR between males and females. For example, total PA days was 3.39 for males and 3.36 for females. Do this difference (0.03 day!) have any meaning?

Author Response

The authors would like to kindly thank the reviewer to carry out a 2nd review of the manuscript. 

We have reviewed the comments and respond to each one of them.

Again, I recommend that that the authors should reconsider the analysis of inactive groups. Simply, the question is which is more active males or females? The proportion of PA recommendation was higher in males, but among inactive groups females showed higher PA days. I think that the distribution of PA day should be shown and PA days could be compared for not inactive groups but for all study subjects.

The reviewer suggested producing a distribution of PA days. This is added to an appendix file and reported on L193

Concerning my former comment 2, small OR difference (less than 0.1) means the difference in OR between males and females. For example, total PA days was 3.39 for males and 3.36 for females. Do this difference (0.03 day!) have any meaning?

We are not sure we follow the reviewers comment, as the OR were for the active population not the inactive population. We have modified from the original paper to state inactive girls and boys levels of PA are equivalent. In most other studies, often activity levels among boys are higher than girls. In our study we do not find that difference among the inactive group. Therefore, we hope the reviewer can agree that this is an important finding. In short, no difference between males and females has significant meaning.